# CRISPR Gene Editing of Murine Blood Stem and Progenitor Cells Induces MLL-AF9 Chromosomal Translocation and MLL-AF9 Leukaemogenesis

**DOI:** 10.3390/ijms21124266

**Published:** 2020-06-15

**Authors:** Evgenia Sarrou, Laura Richmond, Ruaidhrí J. Carmody, Brenda Gibson, Karen Keeshan

**Affiliations:** 1Paul O’Gorman Leukaemia Research Centre, Institute of Cancer Sciences, University of Glasgow, Glasgow G12 0YN, UK; t.sarrou@gmail.com (E.S.); Laura.Richmond@glasgow.ac.uk (L.R.); 2Centre for Immunobiology, Institute of Infection, Immunity & Inflammation, College of Medicine, Veterinary and Life Sciences, University of Glasgow, Glasgow G12 8TA, UK; ruaidhri.carmody@glasgow.ac.uk; 3Royal Hospital for Children, Glasgow G51 4TF, UK; Brenda.Gibson@ggc.scot.nhs.uk

**Keywords:** CRISPR/Cas9, chromosomal translocation, mixed lineage leukaemia, tumourigenesis, stem cells

## Abstract

Chromosomal rearrangements of the mixed lineage leukaemia (*MLL*, also known as *KMT2A*) gene on chromosome 11q23 are amongst the most common genetic abnormalities observed in human acute leukaemias. *MLL* rearrangements (*MLLr*) are the most common cytogenetic abnormalities in infant and childhood acute myeloid leukaemia (AML) and acute lymphocytic leukaemia (ALL) and do not normally acquire secondary mutations compared to other leukaemias. To model these leukaemias, we have used clustered regularly interspaced short palindromic repeats (CRISPR)/Cas9 gene editing to induce MLL-AF9 (MA9) chromosomal rearrangements in murine hematopoietic stem and progenitor cell lines and primary cells. By utilizing a dual-single guide RNA (sgRNA) approach targeting the breakpoint cluster region of murine Mll and Af9 equivalent to that in human MA9 rearrangements, we show efficient de novo generation of MA9 fusion product at the DNA and RNA levels in the bulk population. The leukaemic features of MA9-induced disease were observed including increased clonogenicity, enrichment of c-Kit-positive leukaemic stem cells and increased MA9 target gene expression. This approach provided a rapid and reliable means of de novo generation of Mll-Af9 genetic rearrangements in murine haematopoietic stem and progenitor cells (HSPCs), using CRISPR/Cas9 technology to produce a cellular model of MA9 leukaemias which faithfully reproduces many features of the human disease in vitro.

## 1. Introduction

Rearrangements of the mixed lineage leukaemia gene (*MLLr*) are the most common cytogenetic abnormalities found in infant and childhood acute lymphocytic leukaemia (ALL) and acute myeloid leukaemia (AML) [1] and are found in 80% and 35–50% of all paediatric ALL and AML cases respectively [1]. These leukaemias do not typically acquire many secondary mutations [2] and are thought to result from inappropriate repair of DNA double strand breaks (DSBs) via nonhomologous end joining (NHEJ) in haematopoietic cells [3]. As the 8.3-kb breakpoint cluster region of mixed lineage leukaemia (*MLL*) is located between exons 8 and 13 for almost all *MLLr* cases [4], these fusion genes contain the first 8–13 exons of *MLL* and a variable number of exons from the fusion partner genes (FPGs). Up to 135 *MLL* FPGs have been identified and are well described elsewhere [5,6,7]. These include cytoplasmic proteins (AF6, GAS7 and EEN), septins (SEPT2, SEPT5, SEPT6, SEPT9 and SEPT11) and histone acetyltransferases (CBP and P300) as well as the largest subset, the nuclear proteins. This category contains the most common FPGs such as *ENL*, *ELL*, *AF4*, *AF9* and *AF10*, which account for approximately 80% of all MLL fusions [5]. Of these, MLL-AF9 (MA9) t(9;11)(p22;q23) translocations are found in 30% of *MLLr* AML cases [5] irrespective of age and are commonly found in a subset of adult patients with therapy-related AML [8].

Although the precise functional properties of AF9 are unknown, it has been demonstrated to form multimer complexes with members of the polycomb repressive complex 1 (PRC1) complex and disruptor of telomere silencing (*DOT1L*) via a conserved ANC1 homology domain (AHD), indicating a role in transcriptional regulation [9,10]. These properties appear to play a critical role in leukaemic transformation [11,12]. Oncogenic properties of MLL fusion proteins include interference with transcriptional elongation, resulting in dysregulation of target genes [13,14]. Aberrant expression of MLL target genes is strongly dependent on DOT1L-mediated H3K79 methylation [15], which is a critical requirement for MA9 leukaemogensis [16]. Members of the Homeobox (*HOX*) family are commonly overexpressed in *MLLr* leukaemias as a direct target of MLL fusion proteins [4,17]. The HOX family are critical factors in the self-renewing properties of haematopoietic stem cells (HSCs) and their overexpression results in a differentiation block and an increase in self-renewal of immature myeloid progenitor cells. The *HOX* cofactors *MEIS1* [18,19] and *PBX3* [20] are also direct targets of MLL fusion proteins [21] frequently found upregulated in coordination with HOX targets. Importantly, there is evidence to suggest that MA9-mediated leukaemia can progress even in the absence of *HOX* genes such as *HOXA9* [22], indicating an important role of non-*HOX* family genes. This includes *CDK6* [23], *EVI-1* [24,25] and *EYA1* [26,27], which are also found upregulated in *MLLr* leukaemias as a result of direct binding of MLL-FPG proteins to gene promoters [26,28,29], and play a vital role in leukaemic transformation including cellular immortalization [26], hyperproliferation [23,24], chemoresistance [24] and dysregulated self-renewal [25,27]. CDK6 specifically has been highlighted as a critical effector of *MLLr* leukemogenesis as its depletion in mice with MA9-driven AML was shown to overcome the myeloid differentiation block and to prolong survival in vivo [29].

Numerous murine models of *MLLr* have been generated and have each offered valuable insights. Limitations with murine models are in large part due to the technology used to mimic *MLL* chromosomal rearrangements. An early knock-in model [30] utilized homologous recombination (HR) to generate an *Mll-Af9* fusion gene and displayed a myeloproliferative disorder (MPD), leading primarily to AML characterized by expansion of immature myeloid populations with a small percentage developing B-cell ALL (B-ALL) [31]; thus, this model resembles the phenotypic heterogeneity of *MLLr* leukaemias, which can manifest as AML; ALL; or in a minority of cases, mixed phenotype acute leukaemia (MPAL). However, this system lacked tissue-specific control of MA9 expression throughout development and mice were prone to developmental defects as a result of heterozygosity for wild type (WT) *Mll* [32]. To gain more insight into cell-type specific effects, a subsequent approach involved enrichment of haematopoietic populations from these HR-generated MA9 knock-in mice, followed by secondary transplantation into WT recipients [33]. This also demonstrated efficient generation of AML, especially when Lin−Sca-1+c-Kit+ (LSK) cells, which include HSCs and haematopoietic stem and progenitor (HSPCs), were transplanted. An advantage of these methods is the expression of the MA9 transgene from the endogenous *Mll* promoter, which results in physiological transgene expression levels. This model cannot however address the leukaemia stem cell origin with respect to de novo AML initiation.

Translocator models utilizing the Cre-loxP system have generated MA9 translocations via loxP sites inserted into those intronic regions of endogenous *Mll* and *Af9* genes where breakpoints are most frequently found in human *MLLr* patients [34]. Using Lmo2-Cre to express Cre recombinase, Mll-Af9 was generated in pluripotent stem cells and led exclusively to myeloid leukaemia whereas T-cell-restricted expression using lck-Cre did not lead to leukomogeneis [35]. The advantage of this Cre-loxP system is that MA9 expression can be driven by lineage-restricted promoters from the endogenous loci [35], reflecting physiological expression. Further models targeting Mll-Af9 expression to specific cells within the haematopoietic system have utilized retrovirus-driven expression followed by transduction and transplantation approaches. These methods benefit from its speed and ease of use [36,37] and has led to the identification of both the HSC and the granulocyte and macrophage progenitor (GMP) as potential leukaemic stem cells-of-origin with divergent clinical features [38]. Retrovirus-mediated expression models however drive transgene expressions that are not physiological. To gain greater control over the level of cell-restricted transgene expression, Dox-inducible transgenic models have since been developed [39,40] enabling close-to-physiological levels and reversible transgene expression in a Dox dose-dependent manner. Using this model, it was demonstrated that long-term HSC (LT-HSC) populations result in an invasive and chemoresistant AML with a primitive progenitor phenotype and a distinct stemness-related gene expression pattern [39]. Dox-inducible transgene expression is however not controlled by an endogenous promoter. “Leaky” Cre expression from tissue-specific promoters together with the expensive and time-consuming nature of tissue-restricted strain generation are major limitations with the Cre-loxP and Dox-inducible transgenic model systems. More recent attempts to circumvent some of the issues highlighted above have utilized transcription activator like effector nucleases (TALEN) technology to generate endogenous MA9 [41,42]. These cells exhibited a significantly higher clonogenic potential with colony morphologies consistent with an immature cell type and development of AML, ALL and MPAL upon xenotransplantation [41].

Clustered regularly interspaced short palindromic repeats (CRISPR)/Cas9 technology facilitates creation of DSBs at almost any genomic sequence of interest technology and has revolutionised the way in which we are able to manipulate the genome, affording a more flexible and readily applicable method for generating novel disease models, including chromosomal translocations, with an ease and simplicity far outweighing prior technologies such as TALEN and Cre-LoxP. CRISPR technology has facilitated generation of chromosomal deletions [43] and rearrangements, including inversions [44] and translocations [45,46,47,48]. Here, we demonstrate efficient CRISPR-mediated generation of an endogenous *Mll-Af9* reciprocal translocation in a murine cell line and primary HSPCs. By utilizing a dual-single guide RNA (sgRNA) approach to target the most common breakpoints in human MA9 rearrangements, we show rapid, reliable generation of simultaneous breakpoints and a fusion product at the DNA and RNA levels. The resultant transformed cells reliably recapitulated features of human *MLLr* leukaemias. These include increased expression of MA9 target genes, reduced IL3 dependency, higher clonogenic potential and self-renewal in primary murine HSPCs. This multiplexed CRISPR technology offers a robust and malleable approach for generating murine models of the multitude of human translocation-associated leukaemias.

## 2. Results

### 2.1. A CRISPR/Cas9 System for Mll-Af9 Genome Editing (302)

We utilized a CRISPR/Cas9 system for the generation of murine t(4;9) MA9 chromosomal translocations in murine HSPCs, utilizing simultaneous targeting of the *Mll* and *Af9* genes by CRISPR sgRNAs directed to chromosomes. Based on prior investigation of common *MLL* and *AF9* breakpoint regions in AML patient samples [49] and murine models of MA9 [35], we designed guides targeting frequently translocated intronic regions between *Mll* exons 8 and 9 (Mll sg1 and 2) and exons 10 and 11 (Mll sg3 and 4) and between *Af9* exons 8 and 9 (*Af9* sg1, 2 and 3) (Figure 1A) to generate two potential MLL-AF9 fusion products (Figure 1B). Guide sequences were generated using the CRISPR.MIT web tool (http://crispr.mit.edu) [50,51], which scores each sequence based on a combination of on- and off-target potentials. We selected those guides with the highest scores, indicating an appropriate efficiency of on-target cleavage with minimal off-target potential. Single guide sequences were cloned into a lentiviral CRISPR-green fluorescent protein (GFP) vector (Appendix A), and the resultant high-titre lentivirus was used to transduce the murine HSPC myeloid progenitor 32D cell line, which is IL3-dependent. At day 5 post-transduction, cells were fluorescent activated cell sorting (FACS) sorted for GFP expression and indel formation in GFP^+^ populations were assessed by a surveyor assay (Figure 1C).

Of those guides targeting *Mll* introns, Mll sg1-3 showed approximately equal indel efficiencies of 31–38%, with no cleavage seen using sg4 (Figure 1C and Appendix A). Similarly, *Af9*-targeting sgRNA1 and -3 showed approximately equal efficiencies of 28–37%, while sg2 gave a significantly poorer indel frequency of 7% (Figure 1C). These data demonstrate efficient cleavage of *Mll* and *Af9* loci using a CRISPR system detectable in the bulk population. Based on these results, Mll sg2 was selected for targeting of *Mll* introns 8 and 9, sg3 was selected for introns 10 and 11, and Af9 sg1 and –3 were selected for *Af9* introns 8 and 9.

### 2.2. Endogenous Generation of MA9 Translocation in Murine HSPCs (368)

To generate MLL-AF9 translocations, simultaneous DNA DSBs in the *Mll* and *Af9* genes of murine 32D HSPCs were created (Figure 2A). Cells were transduced with a CRISPR lentivirus targeting either the *Mll* or *Af9* loci or both in combination and were FACS sorted for GFP at day 5 post-transduction. In 32D murine HSPCs, GFP expression > 20% was noted with each combination of dual guides (Figure 2B) and GFP^+^ populations were sorted for verification of the MA9 translocation and downstream functional analysis.

Verification of the MA9 translocation was carried out in GFP^+^ 32D cells at both the DNA and transcript levels by utilizing PCR primers flanking the breakpoint region (Figure 2A) followed by Sanger sequencing of PCR amplicons. Unique banding patterns were seen following electrophoretic analysis of PCR products from 3/4 *Mll* and *Af9* guide combinations (Figure 2C and Appendix A), with sg3/sg3 as the only guide combination which did not generate a product. Amplicons as indicated in Figure 2C were further analysed by Sanger sequencing and alignment to reference *Mll* and *Af9* genomic sequences, which identified the sg2/sg3-transduced sample (band b, Figure 2C) as having correct alignment to both murine *Mll* (NCBI gene ID: 214162) and *Af9* (NCBI gene ID: 70122), validating generation of the MA9 translocation at the genomic level (Figure 2D). Sequence alignment of amplicons from samples transduced with alternative guide combinations (Figure 2C, red arrows excluding b) identified matches in exonic and intronic regions of the mouse genome, including a noncoding region of chromosome 14 and the *Pigz* gene on chromosome 16. This may be indicative of a potential off-target effect of these sgRNA combinations, in line with current knowledge regarding the off-target potential of Cas9 cleavage. RT-PCR and subsequent electrophoresis of RNA from the dual sg2/sg3-transduced sample further validated the translocation at the transcript level; utilizing primers flanking the mRNA breakpoint region (Figure 2B), we generated an amplicon unique to this sample (Figure 2E) which correctly aligned to murine *Mll* (NM_001357549.1) and *Af9* (NM_027326.3) reference transcripts, thus supporting formation of the MA9 mRNA fusion (Figure 2F). Taken together, these data indicate that CRISPR/Cas9-mediated genome editing can be used to induce formation of a murine t(4;9) MA9 translocation in 32D HSPCs and that this fusion gene product is transcribed appropriately.

### 2.3. CRISPR-Mediated MA9 Translocations Recapitulate Features of MA9 Leukaemia In Vitro (622)

Multifactorial functional analysis was carried out firstly on MA9-validated 32D murine HSPCs, in which we examined proliferative capability, IL3-dependency and expression of MA9-target genes. Over 30 days in liquid culture, we saw no significant difference in proliferation between 32D cells transduced with the sg2/sg3 guide combination compared to those transduced with either single guide (Figure 3A), indicating that the MA9 translocation does not confer a demonstrable proliferative advantage in this already immortalised cell line. MA9 leukemic cells have previously shown only moderate IL3 dependency in contrast to 32D cells which are strictly IL3 dependent [52]. Thus, we sought to examine the effects of IL3 starvation on MA9 32D cells in culture. Over 24 h of IL3 starvation, we noted a slight but nonsignificant growth advantage and resistance to starvation-induced stress in the MA9 cells when compared with single guide controls (Figure 3B), suggesting a decrease in IL-3 dependency conferred by CRISPR-generated MA9 translocation in this cell type, consistent with previous data [52]. Analysis of the expression of common MA9-associated genes in sg2/sg3 double-transduced cells revealed significant upregulation of *Dot1l*, *Cdk6* and *Sirt1* mRNA (Figure 3C). Both DOT1l and CDK6 have critical and well-established roles in MLLr function and leukemogenesis [16,29], and *Sirt1* has been shown to be upregulated in response to MA9 expression in murine models [53]. Taken together, these data strongly support that CRISPR-mediated generation of MA9 using the endogenous loci can recapitulate the functional properties of MA9 leukaemic features.

Following verification of the MA9 translocation in 32D murine HSPCs, we next sought to use our approach in primary murine HSPCs. C-Kit^+^ bone marrow (BM) cells were isolated from WT mice and transduced using the validated sg2/sg3 sgRNA singly or in combination. At day 5 post-transduction, cells were sorted for GFP^+^ and expanded in liquid culture for an additional 5 days prior to beginning functional analysis of the translocation in the primary bulk cell population. Equal numbers of cells from liquid cultures were assessed in colony forming cell (CFC) assays to determine self-renewal, proliferation and differentiation capabilities. Following 20 days in a CFC assay, cells transduced with the sg2/sg3 combination of guides displayed significantly increased cell numbers and formed more colonies when compared to single transduced samples (Figure 3D,E). We next carried out immunophenotyping of day 20 CFC colonies in order to assess the myeloid differentiation status of the cells. We saw differences neither in the live/viable population (Figure 3G) nor in the CD11bGr1- or CD11b+F4/80+ myeloid fractions in any of the samples (Figure 3F,I,J), suggesting that the MA9 translocation in this model does not have a negative impact on cell viability or myeloid differentiation. However, the sg2/sg3 MA9 cells showed a significantly larger population of cells expressing the c-Kit progenitor marker (Figure 3F,H), which has been associated with the leukaemia stem cell (LSC) in AMLs including MA9 [36,37]. This highlights not only a proliferative advantage in the double-transduced MA9 population but also an enrichment of a self-renewing LSC-type population. These data indicate that the CRISPR-mediated generation of the MA9 translocation in murine HSPCs mediates leukaemic transformation, indicated by a proliferative advantage and an increase in self-renewal properties.

## 3. Discussion

Herein, we have demonstrated generation of a de novo MA9 translocation in murine HSPCs using state-of-the-art CRISPR/Cas9 technology to create a novel murine leukaemia model which faithfully recapitulates multiple features of human MA9 leukaemia. By rational design of sgRNA flanking breakpoint regions with common occurrence in *MLLr* AML patients and prior MA9 transgenic models, we identified optimal combinations of sgRNA sequences with which to induce simultaneous DSBs at the specific loci. Each guide was validated for individual cleavage efficiency, and subsequently, optimal combinations were identified for multiplexed CRISPR genome editing to induce an MA9 chromosomal translocation in HSPCs. Following CRISPR editing, the presence of a successful MA9 translocation was verified at both the genomic and transcriptional levels by PCR analysis and sequencing. Analysis of proliferation, clonogenicity, differentiation and gene expression robustly demonstrated the functional ways in which this model recapitulates features of the disease in vitro. This novel approach circumvents common caveats of previous methods, with no requirement for transgenic or immunocompromised mice or primary human material. We have demonstrated the ease of use of this approach in cell line and primary murine HSPCs which can be applied to other chromosomal translocations.

Increased clonogenicity and enrichment of progenitor cells with increased self-renewal capacity is a fundamental characteristic of leukaemias with *MLL* rearrangements. Although we saw no proliferative advantage in MA9 32D HSPCs, we did observe a modest reduction in growth factor dependence as previously demonstrated. Normal primary HSPC cells have limited growth potential in vitro, which showed increased colony growth and number when expressing MA9. Increased clonogenicity is associated with increased c-Kit expression, a cell surface marker found in myeloblasts in up to 80% of AML cases [54]. C-Kit expression has also been strongly associated with poor prognoses in other instances of chromosomal translocations such as t(8;21) [55]. We saw a significant enrichment of c-Kit-expressing cells, indicating an enrichment of colony-forming LSC-like early progenitor cells in MA9-expressing cells consistent with leukaemic transformation. Assessment of myeloid differentiation did not reveal a block in expression of differentiation markers in MA9-mediated cells, indicating that, although cells have acquired self-renewal properties, they maintain the ability to differentiate toward the myeloid lineage. A block in terminal myeloid differentiation cannot be ruled out, and whether cells can complete myeloid differentiation remains to be investigated. Cells that are unable to differentiate to the GMP stage of differentiation in murine models of *MLLr* do not undergo leukaemic transformation [56], and so, it is understood that some level of differentiation down the myeloid pathway is required for *MLLr* AML transformation. Data produced by Stavropoulou et al. utilizing a Dox-inducible MA9 model show that surface marker analysis of colonies derived from iMLL-AF9 LT-HSC and GMP subpopulations showed cells expressing the Mac-1 and Gr-1 differentiation markers in addition to c-Kit in the presence of Dox [39].

MA9-initiated oncogenesis has been shown to depend on MA9 modification of transcriptional elongation, resulting in dysregulated expression of target genes. We demonstrated an increase in expression of *Dot1* and *Cdk6*, both of which are essential for the oncogenic function of MA9-rearranged cells [29,57]. Interestingly, we also noted a significant increase in expression of *Sirt1*, which is found significantly overexpressed at the protein level in LSC populations from AML patients [53], where it acts to maintain their survival, growth and drug resistance. The lack of increased *HoxA9* expression observed in our cells may be due to low frequency of translocation within the bulk population analysed, as has been previously noted in TALEN-focused studies [41,42] and CRISPR-generated models [58].

Recently, work by Jeong et al. [58] similarly demonstrated efficient generation of MA9 translocations by CRISPR/Cas9 technology, utilizing methodologies in line with those described herein. The authors used the human system to generate MA9 translocation in human CD34+ cord blood (CB) cells in contrast to the murine system we have presented here. Consistent with our results, human MA9-expressing CB cells in that study demonstrated a growth advantage in liquid culture and clonal expansion in CFC analysis and induction of AML, ALL and MPAL following transplantation in immunocompromised mice. The translocation efficiency achieved by Jeong et al. [58] is considerably higher than in previous TALEN-generated models of MA9 at around 0.5–1%. However, it is important to note that this efficiency remains low overall and so should be taken into consideration when analysing the bulk population. In contrast to the single-cell clonal analysis performed by Jeong et al. [58], we are able to demonstrate the leukaemic transformation effects of CRISPR-generated MA9 in the bulk population that may be masking some effects such as increased Hox gene expression. A considerable advantage of our methodology which allows effects to be seen in the bulk population is the reduction in time in culture of primary HSPCs that is required with single-clone selection. Further single-cell analysis from bulk populations in both the 32D cell line and primary HSPCS would give further insight into individual aspects of MA9-mediated and context-specific gene regulation. A limitation with using primary human HSPCs is the requirement for immunocompromised mice for xenografts. Our work is the first to show the use of this technology to generate the murine MA9 translocation from the endogenous loci faithfully recapitulating MA9-mediated transformation in murine HSPCs. A significant advantage of the murine system is the potential for investigation into immune-, gene- and cell-specific contexts using the myriad of murine genetic models and xenotransplantation approaches.

Our methodology has several advantages for use in genetic models. The tractability of this system is such that we can readily study MA9 translocations in combination with other genetic or epigenetic abnormalities. Generation of new sgRNAs targeting sequences of interest are quickly and easily cloned into the CRISPR vector validated in this work and can be multiplexed as required. Our sgRNA target sequences were also selected not only for their commonality within *MLLr* patients but also for their prior use in alterative murine models of MA9, and as such, our approach offers not only a robust model of human disease but also a means of comparison to alterative methodologies [35,41,42]. The work presented here and by others [48,58] exemplifies the ongoing shift towards precise genome editing with CRISPR technologies to quickly and accurately recapitulate human disease in model systems. We have demonstrated the tractability and potential of this methodology in modelling human MA9 leukaemias, which represents an important advancement in the study of MA9-driven leukaemia.

## 4. Materials and Methods

### 4.1. Cell Culture

All cell cultures were grown at 37 °C, 5% CO_2_. Human embryonic kidney (HEK)-239T and murine embryonic fibroblast NIH-3T3 cells were maintained in Dulbecco Modified Eagle Medium (DMEM; Gibco, Paisley, UK) supplemented with 10% heat-inactivated foetal bovine serum (FBS; Gibco, Paisley, UK), 2 mM L-glutamine (Gibco, Paisley, UK) and 1% Penicillin/ Streptomycin (Pen/Strep; Gibco, Paisley, UK). 32D murine myeloid progenitor cell line HSPCs were maintained in Roswell Park Memorial Institute-1640 (RPMI-1640; Gibco, Paisley, UK) supplemented with 10% FBS, 10% WEHI-3B conditioned media, 2 mM L-glutamine and 1% Pen/Strep. Primary murine HSPCs were cultured in DMEM pre-stimulation media supplemented with 15 % FBS, 2 mM L-glutamine, 1% Pen/Strep, 10 ng/mL recombinant murine interleukin-3 (rmIL3; PeproTech, London, UK), 10 ng/mL recombinant murine interleukin-6 (rmIL6; PeproTech, London, UK) and 100 ng/mL recombinant murine stem cell factor (rmSCF; PeproTech, London, UK) immediately following isolation and were transferred to DMEM activation media supplemented with 15% FBS, 2 mM L-glutamine, 1% Pen/Strep, 10 ng/mL rmIL3, 10 ng/mL rmIL6, 100 ng/mL rmSCF and 4 µg/mL polybrene (Sigma-Aldrich, Dorset, UK) for transduction.

### 4.2. Murine Bone Marrow (BM) Harvest and c-Kit Enrichment

C57BL/6 mice were purchased from Charles River UK and housed in the Beatson Biological Service and Research Units. All mouse experiments were approved by the local animal welfare ethical review body (AWERB) committee and United Kingdom (UK) home office and performed according to UK Home Office project license 60/4512 approved 25 April 2013(Animal Scientific Procedures Act 1986) guidelines. Femur, tibia and pelvic girdle of wild-type (WT) C57BL/6 mice were crushed in phosphate buffered saline (PBS)/2% FBS and passed through a 40-μM cell strainer (Fisher Scientific, Loughborough, UK) to harvest BM. C-kit enrichment was carried out by magnetic activated cell sorting (MACS) separation of BM samples following incubation with anti-CD117 microbeads (Miltenyi Biotech, Bergisch Gladbach, Germany).

### 4.3. sgRNA Design, Synthesis and CRISPR Transduction

Reference mRNA transcripts of genes of interest (http://www.ncbi.nlm.nih.gov/nucleotide/) were used for identification of suitable protospacer sequences using the CRISPR MIT tool (previously available at http://crispr.mit.edu; accessed August 2015) [50,51]. Top and bottom strand protospacers were purchased as single stranded DNA (ssDNA) oligos (IDT, Surrey, UK) (Appendix A) and cloned into pL.CRISPR.EFS.GFP (Appendix A). pL-CRISPR.EFS.GFP was a gift from Benjamin Ebert (Addgene plasmid # 57818) [59]. The all-in-one CRISPR lentivirus was used to constitutively express sgRNAs and Cas9 under the control of human U6 (hU6) and elongation factor 1α (EF-1α) promotors, respectively. HEK-293T and NIH-3T3 cells were used for lentiviral production and titration, respectively, and 32D or primary murine HSPCs were transduced with high-titre CRISPR lentivirus as previously described [60].

### 4.4. Flow Cytometry and Cell Sorting

Samples were stained and analysed in PBS or PBS/2% FBS or sorted for GFP in the same buffer. Single cells were gated by forward scatter area (FSC-A) vs height (FSC-H), and then viable cells were gated by FSC-A vs side scatter area (SSC-A). For viability staining, analysis was performed only in single-cell gates and dead cells were excluded by 4′,6-diamidino-2-phenylindole (DAPI; Sigma-Aldrich, Dorset, UK) staining. Compensation was performed using unstained cells, single stained cells and single stained UltraComp eBeads (Thermo Scientific, Paisley, UK). A complete list of antibodies and dyes can be found in Appendix A.

### 4.5. Surveyor Assay

Genomic DNA was extracted using QuickExtract™ DNA Extraction Solution (Epicentre Biotechnologies, Madison, WI, USA), and analysis with Surveyor Nuclease S was used to calculate indel frequency. Full methods and primer sequences can be found in the Appendix A.

### 4.6. Genomic Target Amplification (GTA)

Genomic DNA was extracted from GFP-sorted cells (day 5 post-sorting) using Epicentre QuickExtract™ DNA Extraction Solution. PCR amplification of target loci was performed using Q5 High-Fidelity DNA Polymerase (New England Biolabs; NEB, Herts, UK), and the products were analysed on 2% agarose (Sigma-Aldrich, Dorset, UK) gel. Bands of interest were excised using the GeneJET Gel Extraction and DNA Cleanup Micro Kit (Thermo Scientific, Paisley, UK) before analysis by Sanger sequencing. Primer sequences can be found in Appendix A.

### 4.7. Reverse Transcription PCR

Total RNA was isolated from GFP-sorted cells (day 5 post-sorting) by either Arcturus PicoPure RNA Isolation Kit (Thermo Scientific, Paisley, UK) or RNeasy Mini Kit (QIAGEN, Manchester, UK)), and target-specific cDNA synthesis was carried out by Superscript III (Thermo Scientific, Paisley, UK) using primers specific for locus of interest. cDNA was amplified using Taq DNA polymerase (NEB, Herts, UK) and target-specific primers. Purified products were analysed on 2% agarose gel, and bands of interest were excised and analysed by Sanger sequencing. Primer sequences can be found in Appendix A.

### 4.8. Gene Expression Analysis (qRT-PCR)

Total RNA was isolated from GFP-sorted cells, and gene expression analysis was carried out as previously described [60]. Results were calculated by the ΔΔCt method with normalization to the *Gusb* housekeeping gene. Primer sequences can be found in Appendix A.

### 4.9. Colony-Forming Cell (CFC) Assay

Methocult GF M3434 (Stem Cell Technologies, Vancouver, BC, Canada) methylcellulose media containing recombinant murine stem cell factor (rmSCF), recombinant murine interleukin-3 (rmIL-3), recombinant human interleukin-6 (rhIL-6) and recombinant human erythropoietin (rhEPO) was used. Equal numbers of BM c-kit-enriched, GFP-sorted cells were prepared in triplicate and resuspended in culture media. Cell suspension was added to methylcellulose and transferred to 3 × 35 mm dishes (Greiner-Bio-One, Gloucestershire, UK) for 10–20 days. Colonies were scored, and cells were analysed by flow cytometry. If required, cells were replated to a second round of CFC following the same protocol.

### 4.10. Statistics

GraphPad Prism (version 5.00.288; LA Jolla, CA, USA) was used for statistical analysis and graphing. Statistical significance was determined by one-way ANOVA with Bonferroni posttest.

## Figures and Tables

**Figure 1 ijms-21-04266-f001:**
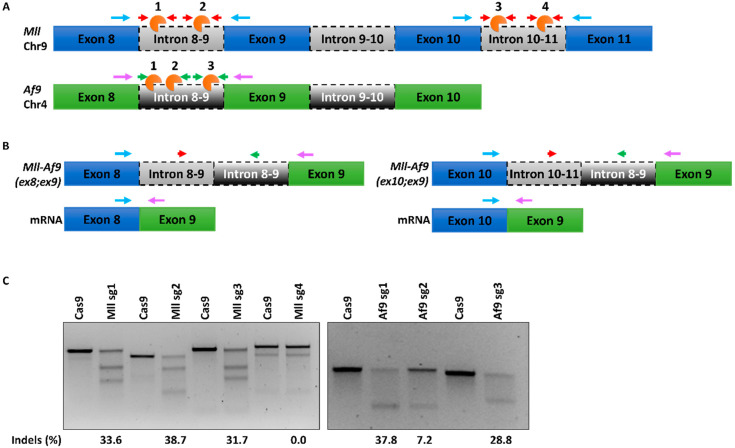
Design and validation of clustered regularly interspaced short palindromic repeats (CRISPR)/Cas9 gene editing tools targeting murine mixed lineage leukaemia (*Mll*) and *Af9* genes: (**A**) Schematic representation of partial *Mll* and *Af9* genomic regions with sgRNA location. Four guides were designed targeting either introns 8 and 9 (*Mll* sg1 and -2) or introns 10 and 11 (*Mll* sg3 and -4) of the *Mll* gene, and three were designed to target introns 8 and 9 (AF9 sg1–3) of Af9. sgRNA locations/Cas9 cleavage sites are shown in orange. Red (*Mll*) and green (*Af9*) arrows represent PCR primers flanking cleavage sites. Blue and purple arrows represent RT-PCR primers on MLL and AF9 transcripts, respectively. (**B**) Representation of two predicted *Mll-Af9* fusion gene products following editing with either set of *Mll* and *Af9*-targeting sgRNAs. (**C**) Surveyor assay of 32D cells transduced with a CRISPR vector expressing the indicated sgRNAs. PCR was performed on genomic DNA from transfected GFP^+^ cell populations using primers as indicated in Figure 1A. Products were assayed by the surveyor and analysed by 2% gel electrophoresis. Percentage indel formation was calculated by quantification of each DNA band. Cells transduced with an empty CRISPR vector (Cas9 only) were used as the control sample for each.

**Figure 2 ijms-21-04266-f002:**
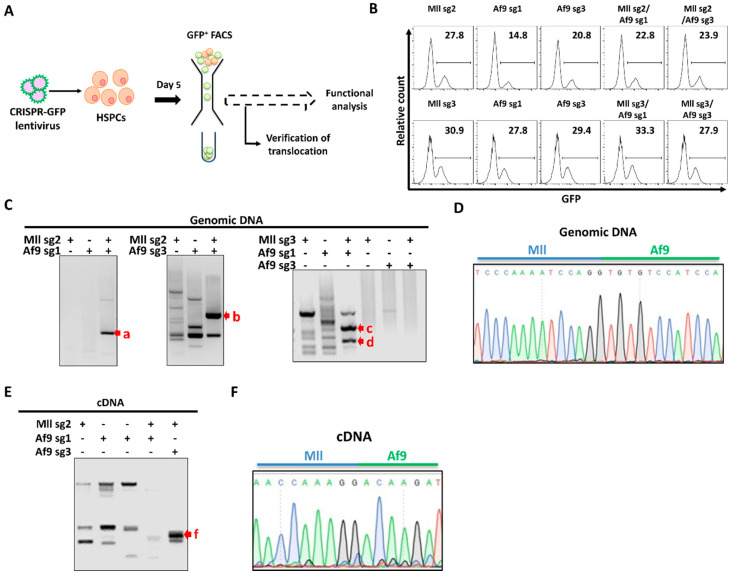
Endogenous generation of MLL-AF9 (MA9) translocation in murine haematopoietic cells: (**A**) Workflow schematic for generation of t(4;9) translocated 32D HSPCs. Cells were transduced with single or dual CRISPR/Cas9 lentivirus and sorted for GFP at day 5 for downstream analysis. (**B**) Flow cytometry histograms showing percentage of GFP^+^ cells at day 5 post-transduction. (**C**) Validation of genomic MA9 translocation: Genomic DNA was extracted from sorted GFP^+^ populations and PCR amplified using primers flanking the MA9 breakpoint. PCR products were then resolved by agarose gel electrophoresis. (**D**) Sanger sequencing of the PCR product from the sg2/sg3 sample (red “**b**” arrow, part **C**): Alignment of the sequence to reference genome verified the alignment with murine *Mll* and *Af9*. (**E**) RT-PCR analysis of the MA9 breakpoint region verifying MA9 translocation at RNA resolution: cDNA was reverse transcribed from RNA and PCR amplified using primers flanking the MA9 breakpoint. (**F**) Sanger sequencing of the sg2/sg3 RT-PCR product: Alignment of the sequence to reference RNA showed alignment to murine *Mll* and *Af9* transcripts.

**Figure 3 ijms-21-04266-f003:**
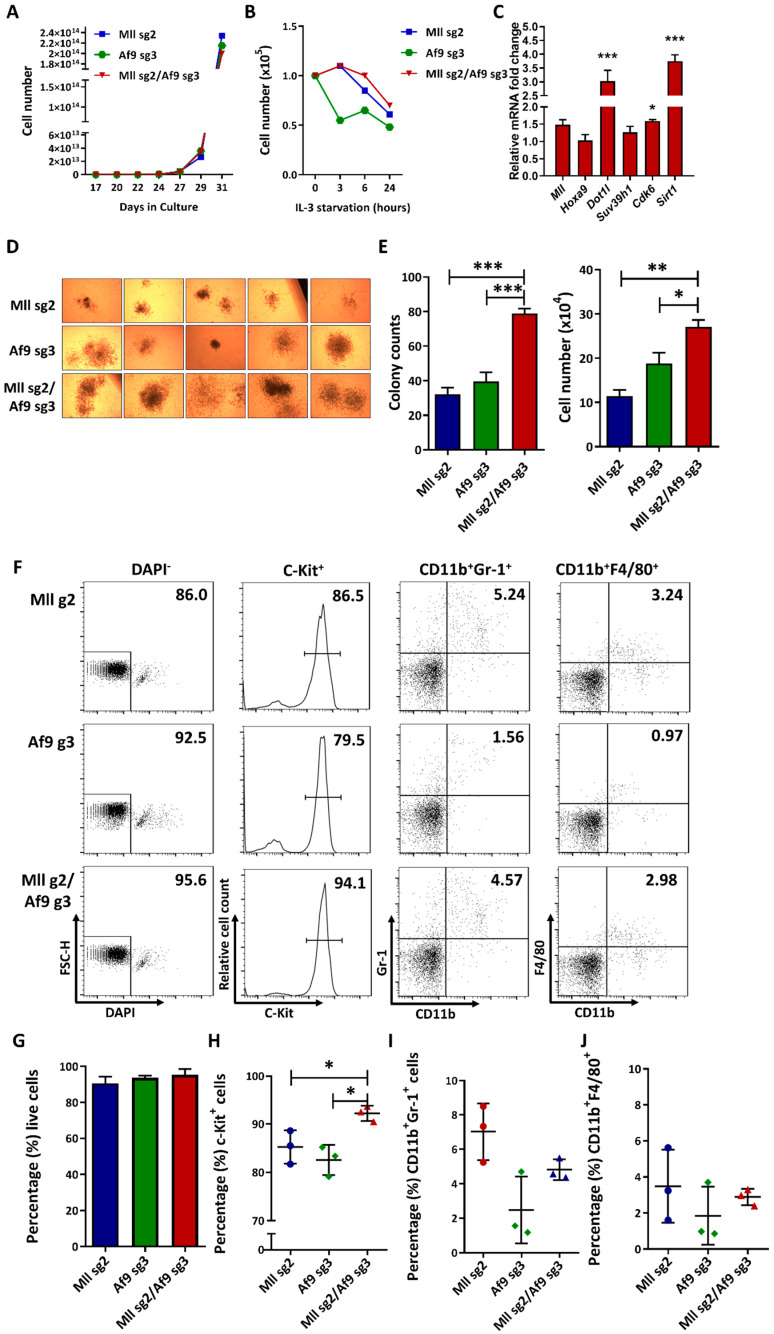
CRISPR-mediated MA9 translocation recapitulates features of MA9 leukaemia: (**A**) Cumulative cell number and growth curve of 32D HSPCs transduced with either *Mll* sg2 or *Af9* sg3 or combination. Cells did not show any significant difference in proliferation following MA9 translocation. (**B**) Cell number and growth curve of 32D HSPCs transduced with *Mll* and/or *Af9* guides as indicated with IL-3 starvation: MA9-translocated cells displayed reduced levels of IL-3 dependency compared to single-guide controls. (**C**) Expression of MA9 target genes: ΔΔCT values of the sg2/sg3 sample are normalised to *Mll* sg2 single guide control. The graph shows mean ± SD of representative experiments. All samples were analysed in technical duplicates. Statistical significance was calculated by one-way ANOVA followed by Bonferroni posttest. *p* < 0.001 is represented as ***, and *p* < 0.05 is represented as *. (**D**) Representative colony pictures (400×) of CFC colonies after ten days in culture (left panel): Samples were plated in technical triplicates. (**E**) Mean ± SD of colony counts (left) and cell numbers (right). (**F**) Representative flow cytometric plots from samples in Figure 3D with the percentage of relevant populations indicated numerically. (**G**–**J**) Percentage of each population in samples shown as (**G**) live (DAPI^-^), (**H**) c-Kit^+^, (**I**) CD11b^+^Gr-1^+^ and (**J**) CD11b^+^F4/80^+^. Each sample is shown as mean ± SD from three technical triplicates. Statistical significance was calculated by one-way ANOVA followed by Bonferroni posttest. *p* < 0.001 is represented as ***, *p* < 0.01 is represented as ** and *p* < 0.05 is represented as *.

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
