# Peer review of "CRISPR Gene Editing of Murine Blood Stem and Progenitor Cells Induces MLL-AF9 Chromosomal Translocation and MLL-AF9 Leukaemogenesis"

_ijms, 2020, doi:10.3390/ijms21124266_

Round 1
Reviewer 1 Report
The manuscript submitted by Sarrou et al entitled: CRISPR gene editing of murine blood stem and progenitor cells induces MLL-AF9 chromosomal translocation and MLL-AF9 leukaemogenesis, is well-written and clearly presented.
The authors have used CRISPR/Cas9 technology to edit the murine genome at two loci simultaneously facilitating the generation of a common translocation seen in Acute Myeloid Leukemia MLL-AF9.
The underlying methods used are not unique nor is the actual translocation, however the combination and twin-targeting has not previously been reported in the murine background.
Major Comments:
It is not clear to this reviewer why serial re-plating of colonies was not done in this study to further examine and quantify the degree of increased self-renewal capacity in the MA9 cells compared to normal primary progenitors.
The approach used results in DSBs and presumably both alleles being involved in the translocation being formed. It is not clear whether this truly reflects the clinical condition and whether retention of an intact MLL gene for instance is required for leukemogenicity.
Minor Comments:
Can reference to Figure A, B, C etc in Results be replaced with Figure 1A,2B,3C etc. as confusing when reading through the manuscript.
What time point was used to examine the MLL-AF9 target genes by QPCR? Could the time point used result in the differential level of expression observed between e.g Hoxa9 and Cdk6.
Could the protein level of MLL-AF9 be determined in the edited cell line?
Does the MLL-AF9 translocation result in extended IL-3 independent growth? If so could this be used to further enrich for the translocation to permit protein/other analysis.
Author Response
Major Comment
1. It is not clear to this reviewer why serial re-plating of colonies was not done in this study to further examine and quantify the degree of increased self-renewal capacity in the MA9 cells compared to normal primary progenitors.
Our Response: Our CFC analysis was carried out after 20 days, which was after replating the primary plates. So our data represents one round of replating which we have simplified to “20 days CFC”.. We did not continue with serial re-plating of the colonies. As the labs are closed due to the COVID-19 pandemic we cannot carry out this analysis. However, we believe that after 20 days in CFC assay we can measurably and significantly conclude that enhanced self-renewal is evident in the double-transduced MA9 population compared to controls.
2. The approach used results in DSBs and presumably both alleles being involved in the translocation being formed. It is not clear whether this truly reflects the clinical condition and whether retention of an intact MLL gene for instance is required for leukemogenicity.
Our Response: As our analysis was on the bulk population, so we cannot conclude if one or both alleles were affected. However, MA9 reciprocal translocation occurs in one allele and there is some evidence that MA9 requires wt MLL (Thiel et al, Cancer Cell, 2010, doi:10.1016/j.ccr.2009.12.034., and Winters & Bernt, Frontiers in Pediatrics 2017, doi: 10.3389/fped.2017.00004).
Minor comment
Our Response:
3. Can reference to Figure A, B, C etc in Results be replaced with Figure 1A,2B,3C etc. as confusing when reading through the manuscrip
Our Response: We provided figure numbers 1A, 2A, etc in the submitted manuscript, that appear to be missing in the reviewed file. We have therefore added back all the numbers to figures throughout – all with tracked changes.
4. What time point was used to examine the MLL-AF9 target genes by QPCR? Could the time point used result in the differential level of expression observed betweeng Hoxa9 and Cdk6.
Our Response: QPR was performed 5 days after GFP positive cell sorting, day 5 post sorting. Line 390 we have now added “ day 5 post-sorting). With regard to the differential expression of genes, we would agree with the reviewer that assessing gene expression, which is a stochastic process, at one timepoint may mask the kinetics of each gene expression changes. Additionally, and most likely is that the gene expression changes occur in population clusters as have been shown by Chen et al, Nature Communications, 2019, doi.org/10.1038/s41467-019-13666-5. Our data is in line with such gene expression changes.
5. Could the protein level of MLL-AF9 be determined in the edited cell line?
Our Response: For protein analysis by western blotting there is no specific fusion protein antibody and the level of detection from the bulk population would be below sensitivity, hence we confirmed fusion protein expression by QPCR transcript detection.
6. Does the MLL-AF9 translocation result in extended IL-3 independent growth? If so could this be used to further enrich for the translocation to permit protein/other analysis.
Our Response: We did not extend the IL-3 starvation beyond the 24 hours for 32D cells. We agree that it could permit enrichment of MA9 expression cells and further analysis such as protein expression. Due to the labs being closed because of the COVID19 pandemic, we cannot carry out this analysis. However, we have confirmed fusion transcript expression so believe we have confirmed the fusion transcript was created in the cells.
We kindly ask that you consider our revised submission for publication.
Kind regards
Reviewer 2 Report
Sarrou et al. demonstrated in this paper that a murine t(4;9) MLL-AF9 (MA9) chromosomal translocation can be generated in 32D cells and c-Kit+ mouse bone marrow cells by using a CRISPR/Cas9 gene editing technology. The authors showed that IL-3 dependency of 32D cells became less stringent in the presence of MA9 gene fusions. In addition, the authors indicated that mouse c-Kit+ bone marrow cells with MA9 had a similar functional property observed in leukemic stem cells in AMLs with MA9. The results of this paper are clear and interesting. However, there are a couple of issues I would like to ask the authors.
- The authors showed the efficiency of generation of MA9 chromosomal rearrangements with this gene editing technology.I wondered whether MA9 generation occurs in one allele or both alleles.
- The authors mentioned in lines 235-236 that “a self-renewing LSC-type population which retain differentiation potential.” But this differentiation potential can be addressed only in in vivo transplantation experiments. In fact, Jeong et al (ref 56) generated MA9 chromosomal translocation in human hematopoietic stem/progenitor cells with a similar gene editing method and examined leukemogenesis from these human cells in immunocompromised mice. The authors should investigate characteristics of c-Kit+ mouse bone marrow cells with MA9 not only in in vitro cultures but also in in vivo injection assays.
- Figure numbers are not properly indicated in the text.Also literatures are not fully referred in some parts (for example, lines 301-332).
Author Response
1. The authors showed the efficiency of generation of MA9 chromosomal rearrangements with this gene
editing technology. I wondered whether MA9 generation occurs in one allele or both alleles.
Our Response: As our analysis was on the bulk population, such analysis would lack sensitivity. However, MA9
reciprocal translocation occurs in one allele and there is some evidence that MA9 requires wt MLL (Thiel et al,
Cancer Cell, 2010, doi:10.1016/j.ccr.2009.12.034., and Winters & Bernt, Frontiers in Pediatrics 2017, doi:
10.3389/fped.2017.00004).
2. The authors mentioned in lines 235-236 that “a self-renewing LSC-type population which retain
differentiation potential.” But this differentiation potential can be addressed only in in vivo
transplantation experiments. In fact, Jeong et al (ref 56) generated MA9 chromosomal translocation in
human hematopoietic stem/progenitor cells with a similar gene editing method and examined
leukemogenesis from these human cells in immunocompromised mice. The authors should investigate
characteristics of c-Kit+ mouse bone marrow cells with MA9 not only in in vitro cultures but also in in
vivo injection assays.
Our Response: We agree with the reviewer that in vivo transplant of the c-Kit+ could provide additional
information on the differentiation potential, while our in vitro data provides information only on the differentiation
state in the culture. We restrict our conclusion to the differentiation state of the cells.
We have therefore deleted “which retain differentiation potential” from the text – line 235-236.
3. Figure numbers are not properly indicated in the text. Also literatures are not fully referred in some
parts (for example, lines 301-332).
Our Response: We provided figure numbers 1A, 2A, etc in the submitted manuscript, that appear to be missing
in the reviewed file. We have therefore added back all the numbers to figures throughout – all with tracked
changes.
We have added ref 56 to the text – line 302, line 309, line 312.
We have added ref 35, 41,42 to text – line 329.
We have added ref 48,56 to text – line 330
Reviewer 3 Report
In this manuscript, Sarrou et al. relied on CRIPSR-CAS9 system to generate endogenous MLL-AF9 gene fusion in myeloblast cell line 32D and murine c-KIT positive HSPCs. The authors showed that they were able to induce the fusion in the cell line as well as in primary c-KIT positive cells. They further went on to validate the biology of fusion construct by clonogenic assays, cell proliferation, and QRT-PCR. The model authors have generated using CRISP/CAS9 is useful to investigate the biology of MLL-fusion leukaemias. However, the authors have carried out all the validating experiments using bulk cells and therefore it is not clear from the data presented whether the model they have generated truly reflects the biology of MLL-AF9 fusion. Literature suggests that the model does represent the biology of MLL-fusion leukaemias using human CD34+ve cells https://www.ncbi.nlm.nih.gov/pmc/articles/PMC6784514/. But authors were not able to show that convincingly in this manuscript.
The authors show that transduction efficiency is less than 20%, so the efficiency of transduction for both the guides would be less than 5%. Ultimately the number of cells carrying MLL-AF9 fusion would be less than 2-3% at the best. While it is clear from the data presented that there are cells carrying this translocation, however, it is not clear what percent of cells are carrying this translocation. I suppose if c-KIT+ cells are cultured for a long time, cells with the translocation may overtake the rest of the cell population. Therefore the conclusion that the fusion construct does not provide any proliferative advantage is not convincing. This is further evident from the lack of induction of some well-known targets of MLL-AF9. The conclusions made based on Figure 3B are again not convincing. There are no stats.
In Figure 3H, how is c-KIT expression maintained after 20 days of culture? Please show Y-axis from 0. An increase in 5-10% may be statistically significant but biologically not meaningful.
Minor points:
32D is a myeloblast cell line, referring the line as murine HSPC cells is confusing.
Figures were not referenced properly (Figure 2a not just Figure A).
It’s not clear when GTA was performed from the methods. Was it done immediately after GFP positive cell sorting or were the GFP positive cells cultured before gDNA was isolated?
Author Response
1.The authors have carried out all the validating experiments using bulk cells and therefore it is not clear from the data presented whether the model they have generated truly reflects the biology of MLL-AF9 fusion. Literature suggests that the model does represent the biology of MLL-fusion leukaemias using human CD34+ve cells https://www.ncbi.nlm.nih.gov/pmc/articles/PMC6784514/. But authors were not able to show that convincingly in this manuscript.
Our response: We have stated that our model recapitulates many features of the disease as we acknowledge not all features were assessed and were limited to bulk cells and in vitro analysis. Our in vitro assessment recapitulates the known in vitro features only of MA9 fusion leukaemias (PMC6784514 cited by the reviewer, ref 56).
2. The authors show that transduction efficiency is less than 20%, so the efficiency of transduction for both the guides would be less than 5%. Ultimately the number of cells carrying MLL-AF9 fusion would be less than 2-3% at the bes While it is clear from the data presented that there are cells carrying this translocation, however, it is not clear what percent of cells are carrying this translocation. I suppose if c-KIT+ cells are cultured for a long time, cells with the translocation may overtake the rest of the cell population. Therefore the conclusion that the fusion construct does not provide any proliferative advantage is not convincing. This is further evident from the lack of induction of some well-known targets of MLL-AF9. The conclusions made based on Figure 3B are again not convincing. There are no stats.
Our response: Our analysis was carried out in bulk cells and therefore without single cell analysis which we cannot provide, we cannot conclude what percent of cells are carrying the translocation. We concluded that there was no demonstrable proliferative advantage in the already immortalised 32D cell line at the bulk population level. We did convincingly demonstrate that the fusion does provide a proliferative advantage in primary HSPCs (figure 3D-E). We have now added a line to address the reviewers comment regarding figure 3A-C: Line 316 in discussion “in both 32D cell line and primary HSPCS” added. With regard to figure 3B, we conclude there is “a slight growth advantage and resistance to….”. We now add “but non-significant” to line 210 as we performed a one-way anova repeated measurements and Friedman non-parametric test followed by Dunns post-test and a two - way anova followed by Bonferroni post-test and both tests returned non-significant.
3. In Figure 3H, how is c-KIT expression maintained after 20 days of culture? Please show Y-axis from 0. An increase in 5-10% may be statistically significant but biologically not meaningful.
Our response: Figure 3H Y- axis has been changed to 0 with a split axis. Data is figure 3H is a CFC assay and the media in these assays enables the maintenance of progenitor-type cells, which would include the c-KIT expressing cells.
Minor points:
1. 32D is a myeloblast cell line, referring the line as murine HSPC cells is confusing.
Our response: As 32D cells are diploid myeloid progenitor cells, we aimed to avoid using the term blasts to avoid confusion with the AML myeloblast. We have clarified that 32D as a HSPC myeloid progenitor in the text (line 136), and referred to them as the cell line to separate from the primary HSPCs. “32D” was added line 161 and 185. Line 190, line 242, line 244 (figure legends).
2. Figures were not referenced properly (Figure 2a not just Figure A).
Our Response: We provided figure numbers 1A, 2A, etc in the submitted manuscript, that appear to be missing in the reviewed file. We have therefore added back all the numbers to figures throughout – all with tracked changes.
3. It’s not clear when GTA was performed from the methods. Was it done immediately after GFP positive cell sorting or were the GFP positive cells cultured before gDNA was isolated?
Our Response: GTA was performed 5 days after GFP positive cell sorting. “day 5 post-sorting” added to methods line 384.
Round 2
Reviewer 2 Report
The authors properly revised the manuscript in this version. Therefore, I believe that this paper is appropriate for publication.
Reviewer 3 Report
All the analyses were done on bulk cells which remains a major limitation of the study.